# Exposure to substance and current substance among school-going adolescents in Timor-Leste

Abigail Esinam Adade[1], Kenneth Owusu Ansah[1,2]*, Nutifafa Eugene Yaw Dey[1], Francis Arthur-Holmes[3], Henry Ofori Duah[4], Agbadi Pascal[3]

**1** Department of Psychology, University of Ghana, Legon, Ghana, **2** Department of Psychiatry, Komfo Anokye Teaching Hospital, Kumasi, Ghana, **3** Department of Sociology and Social Policy, Lingnan University, Tuen Mun, Hong Kong, **4** Research Department, FOCOS Orthopaedic Hospital, Accra, Ghana

* kinetic55@hotmail.com

**Data Availability Statement:** The dataset used in this study is freely available and accessible to the public on the WHO website at https://www.cdc.gov/gshs/countries/seasian/timor_leste.htm.

## Abstract

Few studies have examined how exposure to substance influences adolescent's use of substance in Timor-Leste. We assessed this relationship using nationally representative data from Timor-Leste to address this gap. Data was pulled from the 2015 Timor-Leste Global school-based student health survey. Data of students aged 13-17years (N = 3700) from class 7–11 across schools in Timor-Leste were analyzed for this study. Second-hand smoking exposure (AOR = 1.57 [1.31, 1.89] and parental tobacco use, AOR = 1.94 [1.54, 2.44]) was significantly related to in-school adolescent's current use of substance after adjusting for covariates. Current substance use was also positively associated with being male, being in class 10–12, and being food insecure and negatively associated with having at least three close friends and benefiting from parental supervision. To reduce substance use among in-school adolescents, policymakers must consider the inclusion of all models in the social learning environment of adolescents in Timor-Leste.

## 1 Introduction

Substance use remains a crisis affecting both adults and adolescents [1]. The burden of substance use is more severe among adolescents because they are at a higher risk of using and experimenting with a variety of psychoactive substances [2, 3]. Adolescents commonly use cannabis, alcohol, and tobacco [4]. Globally, about 13.8 million youths (5.6% of school-going adolescents) have used cannabis [5]. It is estimated that adolescents averagely consume six litres of pure alcohol consumption per year [6]. Recent data report that nine out of ten tobacco smokers started before the age of eighteen, and 24 million of these early smokers were 13–15 years old [7]. The prevalence of current substance use among adolescents in Timor-Leste is equally worrying. Prevalence results from school-based studies indicate cigarettes and/or tobacco usage of 29.7% [8], alcohol usage of 12.5% [9], and 5.4% usage of cannabis [10].

Alleviating adolescents' regular use of one or multiple types of substances is a central feature of international and national public health policy to reduce the ravaging effects of substance use on the social, emotional, and cognitive development of adolescents [11, 12]. Substance use

**Funding:** The authors received no specific funding for this work.

**Competing interests:** The authors declare that they have no conflict of interest.

among adolescents is associated with a myriad of debilitating consequences such as poor academic performance and engagement in societal vices [13], respiratory illness and cardiovascular disease [14], anxiety, depression, personality disorders, and suicidal tendencies [15]. Reliance on substances equally impair cognitive abilities such as memory, attention, verbal ability, and intelligence have also been reported [16]. In the worst scenario, using substances leads to death. From 1999 to 2015, the death rate due to drug overdose among adolescents has more than doubled in 2015 [17].

Due to these high prevalence rates and the damaging effects of substance use, many stakeholders including the government of Timor-Leste have invested in reducing the proliferation and usage of substances. The WHO and the United Nations in collaboration with the Timor-Leste government have spearheaded the amalgamation and development of recommendations and mechanisms to tackle substance use [18]. This collaboration, which began in 2011, resulted in the crafting of policies such as the display of the health effects of tobacco use on the most visible surfaces of tobacco packages [18]. Despite such measures, the usage of substances, particularly amongst adolescents is on the surge. While it has become imperative for the government of Timor-Leste to introduce more adequate and strict national laws and policies for controlling access to substances by adolescents [19], evidence from recent research examining factors that maintain and raise the prevalence rates of substance usage in adolescence is greatly needed to supplement these laws and policies [20].

Earlier studies from Timor-Leste have identified peer pressure and cigarette marketing activities [21], poor parental supervision and poverty [8], exposure to cigarette advertisements and promotions, parental smoking, close friends smoking and the amount of pocket money [22] as contributors to adolescent substance use. However, recent studies on exposure to substances and substance use amongst Timor-Leste adolescents are lacking. Although Siziya et al. [22] identified exposure factors such as parental smoking and having close friends that smokes as contributors to adolescent substance use; the authors relied on the 2006 Global Youth Tobacco Survey. Again, these authors focused exclusively on adolescents in grades 7–9 and on cigarette, without examining adolescents in upper secondary grade (grades 9–12) as well as other substances like marijuana, alcohol, and tobacco. Considering these limitations, the most recent Timor-Leste Global school-based student health survey from 2015 was used to confirm whether exposure to substance still has an influence on adolescent's substance use.

We situated our hypothesis within Albert Bandura's social learning theory [23] and Ajzen's theory of planned behaviour [24]. Bandura hypothesised that there are both models and learners in every social learning environment. In this social learning context, learners intentionally or unconsciously observe and imitate behaviours from their models [23]. Bandura further emphasized people's behaviours are driven by purposes and goals that are fuelled by their personal beliefs of self-efficacy and goal expectations from their behaviours in a specific social environment [23]. Self-efficacy which is one of the cardinal factors Bandura identified, primarily influences individuals' self-regulation and expected outcomes. When an individual, thus, has a high belief of having the required capability of executing a behaviour successfully, there is a high likelihood the individual will be goal-directed [23]. Similar to Bandura's assertion, Ajzen theory of planned behaviour emphasized that people make logical and reasoned decisions to execute a particular behaviour by evaluating the kind of information available to them [24]. According to Ajzen, behavioural achievement is based on the person's intentions to engage in the behaviour. These intentions are typically influenced by the value the person places on the behaviour, perceived societal pressure, the ease at which the person can perform the behaviour, the views of significant others as well as opportunities available that may help or hinder the behaviour [24]. The decision of an adolescent to engage in a behaviour, such as the intentional use of illicit drugs is predicted by these factors. Overall, both theories emphasize

how personal dispositions such as self-efficacy, influences an individual's attempt to execute a behaviour.

In our study, adults and parents are the models, in-school adolescents are the learners, and substance use is the behaviour. Given that children grow up under the nurture and upbringing of parents in a microenvironment, adolescents can easily learn both positive and negative behaviours from their parents when they have the strong belief that they have a mastery of executing such behaviours [25]. Also, children grow up in communities outside the primary environment of the home where they equally can learn beneficial or harmful behaviours. Most of these adolescents who observe their parents and significant others learn both the positive and the negative behaviours because they perceive that such behaviours are acceptable among their parents and significant others. Studies conducted elsewhere have credited the social learning theory and theory of planned behaviour as one of the ways to understand the link between exposure and adolescent substance use [26–28]. Both theories also intimated the idea that learners may be vulnerable, and this vulnerability can make them adopt unhealthy behaviours. Generally, it is believed that the period of adolescence is marked by sensation seeking and the vulnerability of engaging in risky activities [29, 30]. This vulnerable period due to high sense of self-efficacy and sensation seeking disposition, makes it difficult for adolescents to resist practicing observed smoking behaviours of their parents, peers, and others in their learning environment [31]. Against this backdrop, we aimed to understand the relationship between in-school adolescents' current substance use and exposure to models that use substances.

## 2 Methods

### 2.1 Study design

This study used a cross-sectional secondary dataset involving the 2015 Timor-Leste Global school-based student health survey (GSHS) [7]. The GSHS was created by the World Health Organization (WHO) in 2003 in collaboration with the United Nations' UNICEF, UNESCO, and UNAIDS, receiving technical support from the Centre for Disease Control and Prevention (CDC). The GSHS is a global school-based survey conducted primarily among a nationally representative student sample aged 13–17 years from low-and-middle-income countries [7]. The survey captures data on an extensive variety of variables measured with valid and standardized instruments [7]. These variables include demographics, alcohol and drug use, sexual behaviours that contribute to HIV infection, sexually transmitted infections, unintended pregnancy, tobacco use, violence and unintentional injury, dietary behaviours, hygiene, mental health, and physical activity.

### 2.2 Study sample and sample size

Three-thousand-seven-hundred-and-four (3,704) students participated and completed the Timor-Leste GSHS. These students comprised those who were in Class 7–11 and aged 13–17. Data collection was conducted with a computer scannable answer sheet which was distributed by trained staff during one standard class period. For the sampling technique, the GSHS utilized a two-stage cluster sample design with the mission to generate representative data of all students in Class 7–11. The initial stages of data collection consisted of the selection of schools that reflected the probability proportion of the entire enrolment size in the country. As a result of this school selection process, 38 schools were randomly sampled. At the second stage, classes of the sampled schools were randomly chosen with random start and all students in these classes were eligible to participate. About 4691 students were eligible for data collection. The data collection produced a 100% school response rate, 79% student response, and 79% overall data collection response rate.

Data of 3700 students were used for this study.

## 2.3 Measures

**2.3.1 Outcome variable.**   Adolescent current substance use was treated as the outcome variable in this study. This variable was created by combining four single-item questions measuring students current use of specific substances including cigarettes, marijuana, alcohol, and tobacco. Combination of the four single-item into one variable was based on a similar approach by Moilanen et al. [32]. These set of questions can be found in Table 1. We created our current substance use variable out of the GSHS binary generated versions of these four questions [7]. These GSHS binary variables have a response format of "Yes = 1" and "No = 0" and we kept this response for our newly generated variable.

## 2.3.2 Predictor variable

The predictor variable for this study was exposure to substance. It was conceptualized from two variables namely people smoking in child's presence or second-hand smoking exposure

**Table 1. Weighted socio-demographic characteristics of the of the students.**

| Variable | Frequency (%) | Survey questions, GSHS generated binary and coding |
|---|---|---|
| **Current substance use** | | During the past 30 days, on how many days did you have at least one drink containing alcohol? |
| (0) No | 2394 (63.57) | (0) 0 days |
| (1) Yes | 1306 (36.43) | (1) 1 or 2 days/1 to 2 days/3 to 5 days/6 to 9 days/10 to 19 days/20 to 29 days/All 30 days |
| Total | 3700 (100) | During the past 30 days, on how many days did you smoke cigarettes? |
| | | (0) 0 days |
| | | (1) 1 or 2 days/1 to 2 days/3 to 5 days/6 to 9 days/10 to 19 days/20 to 29 days/All 30 days |
| | | During the past 30 days, how many times have you used marijuana (also called Ganja)? |
| | | (0) 0 days |
| | | (1) 1 or 2times/3 to 9times/10 to 19times/20 or more times |
| | | During the past 30 days, on how many days did you use any tobacco products other than cigarettes, such as Joker, LA, Gudang garam, Sigaru 23, Surya, snuff, chewing tobacco, or betel? |
| | | (0) 0 days |
| | | (1) 1 or 2 days/1 to 2 days/3 to 5 days/6 to 9 days/10 to 19 days/20 to 29 days/All 30 days |
| **People smoke in presence** | | During the past 7 days, on how many days have people smoked in your presence? |
| No | 798 (20.27) | (2) 0 days |
| Yes | 2858 (79.73) | (1) 1 or 2 days/3 or 4 days/5 to 6 days/All 7 days |
| Total | 3656 (100) | |
| **Parental Tobacco Use** | | Which of your parents or guardians use any form of tobacco? |
| No | 2432 (67.64) | (2) Neither/I do not know |
| Yes | 1186 (32.36) | (1) My father or male guardian/My mother or female guardian/both |
| Total | 3618 (100) | |
| **Age** | | How old are you? |
| 11-15yrs | 1758 (40.63) | (0) 11years old or younger-15years old |

*(Continued)*

**Table 1.** (Continued)

| Variable | Frequency (%) | Survey questions, GSHS generated binary and coding |
|---|---|---|
| 16years and above | 1872 (59.37) | (1) 16years old -18 years old or older |
| Total | 3630 (100) | |
| **Gender** | | What is your sex? |
| Male | 1625 (50.68) | (1) Male |
| Female | 1877 (49.32) | (2) Female |
| Total | 3502 (100) | |
| **Grade in school** | | In what grade are you? |
| Class 7–9 lower secondary | 2441 (60.19) | (0) Class 7 (EBC. 3 Ciclo)/Class 8 (EBC. 3 Ciclo)/Class 9 (EBC. 3 Ciclo) |
| Class 10–12 upper secondary | 1155 (39.81) | (1) Class 10 (ES)/Class 11 (ES)/Class 12 (ES) |
| Total | 3596 (100) | |
| **Food insecurity** | | During the past 30 days, how often did you go hungry because there was not enough food in your home? |
| Secure | 3215 (88.27) | (1) Most of the time/Always |
| Insecure | 429 (11.73) | (0) Never/Rarely/Sometimes |
| Total | 3644 (100) | |
| **Number close friends** | | How many close friends do you have? |
| 0 | 172 (4.76) | (1) 0 |
| 1 | 374 (9.70) | (2) 1 |
| 2 | 574 (14.63) | (3) 2 |
| 3+ | 2496 (70.91) | (0) 3 or more |
| Total | 3616 (100) | |
| **Colleague support** | | During the past 30 days, how often were most of the students in your school kind and helpful? |
| No | 2607 (72.27) | (0) Never/Rarely/Sometimes |
| Yes | 960 (27.73) | (1) Most of the time/Always |
| Total | 3567 (100) | |
| **Parental supervision** | | During the past 30 days, how often did your parents or guardians check to see if your homework was done? |
| No | 2607 (71.49) | (0) Never/Rarely/Sometimes |
| Yes | 1022 (28.51) | (1) Most of the time/Always |
| Total | 3629 (100) | |
| **Parental connectedness** | | During the past 30 days, how often did your parents or guardians understand your problems and worries? |
| No | 3219 (88.10) | (0) Never/Rarely/Sometimes |
| Yes | 407 (11.90) | (1) Most of the time/Always |
| Total | 3626 (100) | |
| **Parental bonding** | | During the past 30 days, how often did your parents or guardians really know what you were doing with your free time? |
| No | 2773 (76.43) | (0) Never/Rarely/Sometimes |
| Yes | 23.57 (23.57) | (1) Most of the time/Always |
| Total | 3580 (100) | |
| **Parental respect** | | During the past 30 days, how often did your parents or guardians go through your things without your approval? |
| No | 957 (27.81) | (0) Most of the time/Always |
| Yes | 2646 (72.19) | (1) Never/Rarely/Sometimes |
| Total | 3603 (100) | |

(X1) and parental use of tobacco (X2). Both variables were measured using a single-item question. For X1 the question, "During the past 7 days, on how many days have people smoked in your presence?" was used. Students responded to this question on a 5-point response scale namely, "0 days", "1 or 2days", "3 or 4days", "5 to 6days" and "All 7days". For X2 the question, "Which of your parents or guardians use any form of tobacco?" was used. Students responded to this question also on a 5-point response scale namely "Neither", "My father or male guardian", "My mother or female guardian", "Both" and "I do not know". A binary version of these questions was generated by GSHS to capture the percentage of participants who were exposed and those not bearing a "Yes" (1) and "No" (0) response. We used this binary variable in our analyses.

**2.3.3 Control variables.** Our selection of control variables was motivated by their availability in the dataset and based on evidence from previous school-based research [33, 34]. The control variables are this: Gender, age, grade in school, number of friends, colleague support, food insecurity and parental involvement were the variables selected. More details about the control variables, their respective survey questions and the coding used are presented in Table 1.

## 2.4 Ethics and data accessibility

The World Health Organization's Ethical Committee alongside ministries of education and/or health in each participating country granted ethical approval for the survey. Before the data collection procedure, the enumerators obtained child assent and parental/adult consent. The dataset used in this study is fully anonymized and freely available and accessible to the public on the WHO website at https://www.cdc.gov/gshs/countries/seasian/timor_leste.htm.

## 2.5 Data preparation and analysis

Data analyses began in Stata version 14 by cleaning the data and recoding the variables of interest. We entered the "svyset" command adjusting for clusters, stratification, and sample weights, all performed to account for the complex sampling design contained in the dataset. This procedure is recommended to control for potential analytic errors and allow proper inferences from the data [35]. Univariate analysis was subsequently performed generating frequencies and percentages of the study variables. We next conducted Chi-square test to examine the bivariate relationship between the study variables. We moved on next to conducting multivariable analyses in logistic regression by entering the "logistic" command. Four models of logistic regression analyses were conducted to examine the objective of this study. The first one examined the relationship between people smoke in presence (X1) and adolescent current substance use (Y). The second model examined the relationship between parental Tobacco Use (X2) and Y. Third model predicted X1, X2 onto Y. The last model examined the relationship between X1, X2, Y while controlling for covariates. In all these analyses both crude and adjusted odd ratios were reported.

## 3 Results

### 3.1 Socio-demographic characteristics of students

The prevalence of adolescent current substance use was 36.43% [95% CI:33.2, 39.78]. About 79.73% of the school-going adolescents reported people smoked in their presence and 32.36% reported their parents use some form of tobacco. Greater number of students (59.37%) were aged 16+ years. The sex percentage rate of students was 50.68% males and 49.32% females. About 60.19% were in Class 7–9 whiles 39.81% were in Class 10–12. Adolescents from food

insecure households were approximately 11.77%. Only 4.76% of the students had no close friends. About 27.73% of the students reported having colleague support. In terms of parental involvement, 28.51% of parents assisted their children to complete their homework. About 11.90% of parents understood the problems their children were experiencing and 23.57% had knowledge about where their children spent their free time. About 72.19% of the parents go through their child's things without any approval from the child. See summary of results on Table 1.

### 3.2 Bivariate analysis examining the link between smoking in the presence of adolescents, parental tobacco use, adolescent current substance use substance use and covariates

We performed bivariate analyses with Chi-square test of independence to examine the relationship between people who smoke in the presence of adolescents, parental tobacco use, adolescent substance use and covariates. The results revealed that smoking in the presence of adolescents and parental substance tobacco use were both significantly related to adolescent current substance use. Thus, 38.14% of students who reported people smoke in their presence, also engaged in substance use. Also, 46.67% of the students with parents who smoke tobacco are currently using substance. We further found significant relationships between the covariates and adolescent current substance use which is summarized in Table 2.

### 3.3 Multivariable logistic regression

Table 3 contains the results of logistic regression estimating the association between smoking in the presence of the adolescent and parental tobacco use on adolescent substance use. In Model 1, it was noted that adolescents who were in the presence of those who smoked were more likely to engage in substance use than students who were not in the presence of those who smoked [OR = 1.59, 95% CI: 1.30, 1.94]. In Model 2, it was observed that students with parents who use tobacco were more likely to engage in substance use than those with parents who do not use tobacco [OR = 1.95, 95% CI: 1.53, 2.48]. In Model 3, when both predictors were included, people who smoked in the presence of others were associated with greater odds of adolescent substance use [OR = 1.60, 95% CI: 1.33, 1.93]. It was further noted that, parental tobacco use was associated with greater odds of adolescent substance use [OR = 1.93, 95% CI: 1.51, 2.48]. In Model 4, after adjusting for covariates (gender, age, grade, close friends, colleague support, food insecurity, parental supervision, parental connectedness, parental bonding, and parental respect), both predictor variables remained associated with current substance use. These findings are consistent with our hypotheses that parental substance use and people smoking in the presence of the adolescent, increases the likelihood of adolescent substance use. It was also noted in Model 4 that male students were more likely to engage in substance use than female students. Students in the upper secondary (Class 10 to 12) were more likely to engage in substance use than students in the lower secondary (Class 7 to 9). Students with 3 or more friends were less likely to engage in substance use than students with no friends. Students who were food insecure were more likely to be currently using substance than students who were food secure. Students who experienced parental supervision were less likely to engage in substance use than students who experience no parental supervision.

## 4 Discussion

This study used the 2015 Timor-Leste Global School-based Student Health Survey to examine exposure to substance and substance usage among school-going adolescents. Over 36% of

**Table 2. Chi-Square examining the relationship between adolescent' substance use, independent variables and covariates.**

| Variables | Substance use | | Statistics |
|---|---|---|---|
| | *n* (%) | *n* (%) | |
| | Yes | No | |
| **People smoke in presence** | | | $\chi^2 = 26.64$, p = .0001 |
| No | 221 (27.94%) | 577 (72.06%) | |
| Yes | 1053 (38.14%) | 1803 (61.86%) | |
| **Parental tobacco use** | | | $\chi^2 = 83.95$, p = .000 |
| No | 703 (31.03%) | 1729 (68.97%) | |
| Yes | 556 (46.67%) | 629 (53.33%) | |
| **Age** | | | $\chi^2 = 42.66$, p = .0019 |
| 11–15 | 520 (30.01%) | 1235 (69.99%) | |
| 16+ | 752 (40.63%) | 1119 (59.37%) | |
| **Gender** | | | $\chi^2 = 322.13$, p = .000 |
| Female | 412 (21.12%) | 1464 (78.88%) | |
| Male | 803 (50.23%) | 820 (49.77%) | |
| **Grade in School** | | | $\chi^2 = 37.61$, p = .0007 |
| Class 7–9 | 778 (32.13%) | 1662 (67.87%) | |
| Class 10–12 | 478 (42.17%) | 675 (57.83%) | |
| **No. of close friends** | | | $\chi^2 = 5.94$, p = .0802 |
| 0 | 78 (44.38%) | 93 (55.62%) | |
| 1 | 147 (39.50%) | 226 (60.50%) | |
| 2 | 185 (33.13%) | 389 (66.87%) | |
| 3 or more | 836 (35.09%) | 1658 (64.91%) | |
| **Colleague support** | | | $\chi^2 = .36$, p = .5825 |
| No | 905 (35.95%) | 1701 (64.05%) | |
| Yes | 326 (34.88%) | 633 (65.12%) | |
| **Food insecurity** | | | $\chi^2 = 24.93$, p = .0018 |
| Secure | 1084 (34.91%) | 2129 (65.09%) | |
| Insecure | 198 (47.28%) | 230 (52.72%) | |
| **Parental supervision** | | | $\chi^2 = 5.43$, p = .0164 |
| No | 935 (36.96%) | 1671 (63.04%) | |
| Yes | 321 (32.85%) | 700 (67.15%) | |
| **Parental connectedness** | | | $\chi^2 = 2.55$, p = .2830 |
| No | 1092 (35.38%) | 2126 (64.62%) | |
| Yes | 164 (39.30%) | 243 (60.70%) | |
| **Parental bonding** | | | $\chi^2 = 2.33$, p = .3092 |
| No | 930 (35.27%) | 1842 (64.73%) | |
| Yes | 311 (38.15%) | 496 (61.85%) | |
| **Parental respect** | | | $\chi^2 = 14.94$, p = .0258 |
| No | 394 (41.13%) | 563 (58.87%) | |
| Yes | 863 (34.22%) | 1781 (65.78%) | |

school-going adolescents use alcohol, tobacco, cigarette or marijuana which is consistent with research that found 38% of poly substance use among school-going adolescents [36].

Exposure to substance (being in the presence of smokers or second-hand smoking and parental use of tobacco) was positively related to in-school adolescent's current substance usage. Bandura's social learning theory argues that through exposure and observation, people learn and model others' behaviours [23]. Adolescents in Timor-Leste may have appraised the

**Table 3. Summary of logistic regression examining the relationship between exposure to substance and usage of substance amongst school-going adolescents.**

| | Model 1 | Model 2 | Model 3 | Model 4 |
|---|---|---|---|---|
| Variables | OR [95% CI] | OR [95% CI] | AOR [95% CI] | AOR [95% CI] |
| **People smoke in presence** | | | | |
| No | 1 [ref] | | 1 [ref] | 1 [ref] |
| Yes | 1.59***[1.30, 1.94] | | 1.60*** [1.33, 1.93] | 1.57*** [1.31, 1.89] |
| **Parental tobacco use** | | | | |
| No | | 1 [ref] | 1 [ref] | 1 [ref] |
| Yes | | 1.95*** 1.53, 2.48] | 1.93*** [1.51, 2.48] | 1.94***[1.54, 2.44] |
| **Age** | | | | |
| 11–15 | | | | 1 [ref] |
| 16+ | | | | 1.30 [0.92, 1.82] |
| **Gender** | | | | |
| Female | | | | 1 [ref] |
| Male | | | | 4.13*** [3.16, 5.40] |
| **Grade in School** | | | | |
| Class 7–9 | | | | 1 [ref] |
| Class 10–12 | | | | 1.60** [1.23, 2.09] |
| **No. of close friends** | | | | |
| 0 | | | | 1 [ref] |
| 1 | | | | 0.74 [0.44, 1.25] |
| 2 | | | | 0.58 [0.33, 1.02] |
| 3 or more | | | | 0.59* [0.35, 0.98] |
| **Colleague support** | | | | |
| No | | | | 1 [ref] |
| Yes | | | | 1.00 [0.82, 1.22] |
| **Food insecurity** | | | | |
| Secure | | | | 1 [ref] |
| Insecure | | | | 1.55* [1.09, 2.20] |
| **Parental supervision** | | | | |
| No | | | | 1 [ref] |
| Yes | | | | 0.76** [0.66, 0.89] |
| **Parental connectedness** | | | | |
| No | | | | 1 [ref] |
| Yes | | | | 1.00 [0.67, 1.50] |
| **Parental bonding** | | | | |
| No | | | | 1 [ref] |
| Yes | | | | 1.10 [0.85, 1.41] |
| **Parental respect** | | | | |
| No | | | | 1 [ref] |
| Yes | | | | 0.82 [0.59, 1.15] |
| **More details** | | | | |
| Number of strata | 19 | 19 | 19 | 19 |
| Number of PSUs | 38 | 38 | 38 | 38 |
| Design $df$ | 19 | 19 | 19 | 19 |
| F value | $F(1, 19) = 23.37$ | $F(1, 19) = 33.39$ | $F(2, 18) = 37.36$ | $F(14, 6) = 20.12$ |
| Prob > F | 0.0001 | 0.0000 | 0.0000 | 0.0007 |
| $N$ | 3654 | 3617 | 3588 | 3,040 |
| Population size | 84,473.14 | 83,774.846 | 83,098.112 | 70,383.891 |

(*Continued*)

**Table 3.** (Continued)

|  | Model 1 | Model 2 | Model 3 | Model 4 |
|---|---|---|---|---|
| **Variables** | **OR [95% CI]** | **OR [95% CI]** | **AOR [95% CI]** | **AOR [95% CI]** |
| McKelvey and Zavoina's $R^2$ | 0.196 | 0.406 | 0.483 | 0.853 |

*Note.* CI: 95% confidence intervals

* $p < 0.05$

** $p < 0.01$

*** $p < 0.001$

smoking behaviours around them as favourable [37], something worth trying, and modelled these behaviours. This series of learning processes could have increased susceptibility to initiate smoking and develop the behaviour of substance use. This is consistent with earlier studies suggesting that adolescent's uptake of tobacco smoking was substantially increased by second-hand smoking exposure [22, 38, 39]. The influence of second-hand smoking exposure is even greater when the child is surrounded by multiple smokers including parents, grandparents, older siblings or other people living outside the household [40, 41]. Such levels of exposure to nicotine activates neural pathways that increase the sensitivity of the brain to nicotine and promote the urge to smoke and try out other substances [42]. Our results are further supported and explained by similar studies conducted elsewhere [33, 43]. For instance, Jawad et al. [43] found that that youth waterpipe smoking was significantly influenced by parental tobacco use. In a nutshell, the involvement of one or both parents as smokers have a major effect on adolescents' development of tobacco use behaviours, leading to the intergenerational transmission of smoking behaviours within families [44, 45]. Furthermore, the theory of planned behaviour asserts individuals are more likely to engage in a particular health behaviour if they believe the behaviour will result in specific outcomes that are value. Nicotine, like other drugs, activates reward pathways in the brain-circuitry that regulates reinforcement and pleasurable feelings [46]. Adolescents seek sensation and value pleasure (28). As a result, the pleasurable effects of substances such as nicotine contribute to adolescent substance abuse. In addition, due to the highly addictive properties of nicotine, it is more likely that these exposed adolescents may begin to experiment with other substances and increasing their vulnerability of becoming illicit substance consumers for life [47].

Males were more likely to use substance than females. The variation in use between adolescent males and females in Timor-Leste may be due to biological and social or intrinsic factors. Biologically, males and females exhibit differences in sexual dimorphisms in the brain, endocrine (e.g., ovarian hormones), and metabolic processes, all of which play a significant role in the use and abuse of substance [48]. The consensus in the literature is that males have lesser intoxication rate of alcohol and other substances compared to females due to higher substance related metabolism [48]. As a result, males can consume more substances than their female counterparts. Typical traditional masculine norms expressed through aggression, dominance and risk taking [49] as well as intrinsic factors like high novelty seeking or impulsivity personalities [50, 51] further put males ahead of females in the utilization of substance. All these factors may account for the differences in substance use between boys and girls in Timor-Leste. Our result is consistent with findings among school-going adolescents in Morocco [2, 52], South East Asian countries [53] and Pacific Island countries [34]. Our findings also run parallel with a study by Lev-Ran et al. [54] who revealed a high prevalence of psychoactive drugs (i.e., alcohol, sedatives, cannabis, tranquilizers, opioids, hallucinogens, and cocaine) usage and abuse in men.

Adolescents in upper secondary grade were more likely to currently use substances than those in lower secondary grade. Perhaps adolescents in upper grade may be using substance to cope with the increasing stressors typically associated with upper grade levels [55]. Some of these reported stressors amongst seniors include pressure from studying, academic workload, self-expectation stress and study despondency [56]. Also, upper secondary grade is associated with greater peer control [57]. Due to this great influence, adolescents in upper grades who find themselves in peer groups that use, or abuse substances may be compelled through the sense of group conformity to also use [58]. This is consistent with previous literature indicating high substance usage amongst adolescents in higher grades[52, 59, 60].

Adolescents who were food insecure was more likely to use substance. Food insecurity is the limited or uncertain availability of nutritionally adequate safe food or the inability to purchase food in a socially acceptable manner [61]. Evidence suggest that Timor Leste is a food insecure country with about 36% of households experiencing chronic levels of food insecurity [62]. Given this high chronic prevalence and the 11.73% prevalence found in our study, there is an urgent need for solving food insecurity crisis in Timor-Leste. Nevertheless, it is possible that adolescents who are food insecure are using substance in a way to cope with their situation. A survey among adolescent street children revealed that they use psychoactive substances to alleviate hunger and anxiety as well as stimulants and opioids to suppress their appetite [63]. Likewise, food insecure adolescents may be using multiple substances, which are often more accessible, to alleviate their feelings of hunger. Our finding is consistent with previous studies. One study demonstrated in an urban youth population survey that food insecurity correlates positively with substance use [64]. Another study among a sample of adults in USA showed that food insecurity was associated with smoking cigarettes and heavy alcohol use [65].

School going adolescents with three or more close peers were less likely to use substance than those with no friend. Interestingly, the influence of close peers is either positive or negative. It is argued that many cases of behavioural problems and substance use observed in adolescents is associated with similar substance use behaviours in close friends [66]. On the other hand, consistent with our research, close peer relationships discourage substance usage among adolescents [67]. It is possible that adolescents with more close friends participate in gainful extracurricular activities that influences them to avoid using substance. These activities may include sports, community services and volunteering which have been linked to reduced use of illicit drugs [32, 68]. Such levels of participation may lead to additional benefits including positive health outcomes, increase academic performance, improve moods, and tighter bonds [69].

Finally, children who received parental supervision in the past 30 days were less likely to use substances. Within the microsystem, parents are nearest to the child, and the relationship that takes place in the system typically has a significant effect on the child's life. Parents who regularly monitor their children serve as protective agents against the use of substances as parents keep track of their children' peers, whereabouts, and activities [70]. Moreover, children with good parental supervision have a lower susceptibility to peer pressure which might serve as a contributor to substance use [71]. This is because during the developmental phase, parents are very important in the lives of children and they can influence their children's choices at that time to stay away from risky habits such as substance use [72, 73]. Our finding is consistent with that of Tornay et al's [74] who revealed that substance use decreases as parental regulation increases.

## 5 Implications of the study

The findings of this study call for more stringent interventions and health programs directed towards educating parents about the consequences of using illicit drugs in presence of their

children. To curb the use of substances among school-going adolescents in Timor-Leste certain school-based measures must be implemented. Awareness clubs, for instance, should be formed to educate school-going adolescents about the adverse effects of drug use and abuse. Finally, the policies regarding the use of illegal substances in Timor-Leste should be strengthened to reduce the proliferation and adolescents' access of psychoactive drugs.

## 6 Strengths and limitations

Our study used nationally representative sample collected with robust procedures that increase the generalizability and reliability of the results. Some limitations are worth mentioning. Firstly, the data were collected through self-report questionnaire. Although the World Health Organization's current dataset is the 2015 Timor-Leste Global school-based student health survey, future researchers should try to use current datasets done by other agencies because behaviour changes over time and the 2015 dataset does not entirely reflect current behavioural trends. It is also possible that recall bias and deliberate misreporting affected the accuracy of the data. Additionally, since secondary data was primarily used in this study, we had little control over the variables to be included in our analysis. For instance, we could not operationalize and control for other essential components of Bandura's Social Cognitive Theory such as self-efficacy expectation and outcome expectancy. Despites its importance, we could not control for variables based on Ajzen's theory of planned behaviour. Future studies should use other methodologies to exert more control over variable selection and inclusion. Lastly, we only reported correlational findings and not cause-and-effect since the dataset was collected in a cross-sectional manner. It is recommended that future studies should causally examine these variables using longitudinal research designs.

## 7 Conclusion

This study examined the relationship between exposure to substance and current substance usage among school-going adolescents. We found that parental tobacco use and people smoking in the presence of adolescent or second-hand smoking exposure, increases the likelihood of substance use among adolescents. Furthermore, males, adolescents in the upper secondary, those who were food insecure were more likely to be current substance users whilst adolescents with 3 or more friends and who had parental supervision were less likely. Stakeholders in Timor-Leste should consider these factors in optimizing public health interventions against substance use for adolescents.

## Acknowledgments

We appreciate the WHO for making the dataset used in this study publicly and freely available. We are grateful to the Research Empowerment Network (REN) for the training support in data management and academic writing.

## Author Contributions

**Conceptualization:** Kenneth Owusu Ansah, Nutifafa Eugene Yaw Dey, Henry Ofori Duah, Agbadi Pascal.

**Data curation:** Nutifafa Eugene Yaw Dey, Agbadi Pascal.

**Formal analysis:** Kenneth Owusu Ansah, Nutifafa Eugene Yaw Dey.

**Writing – original draft:** Abigail Esinam Adade, Kenneth Owusu Ansah, Nutifafa Eugene Yaw Dey.

**Writing – review & editing:** Francis Arthur-Holmes, Henry Ofori Duah, Agbadi Pascal.

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
