## [Decision Letter · Decision Letter 0]

6 Dec 2021

PGPH-D-21-00643

Exposure to substance and current substance use of school-going adolescents in Timor-Leste: Evidence from the 2015 Timor-Leste Global school-based student health survey

Dear Mr. Ansah,

Thank you for submitting your manuscript to PLOS Global Public Health. After careful consideration, we feel that it has merit but does not fully meet PLOS Global Public Health’s publication criteria as it currently stands. Therefore, we invite you to submit a revised version of the manuscript that addresses the points raised during the review process.

Kindly carefully consider the issues raised by the reviewers, they are pertinent to contribute to the strength of the argument the paper is making.

We look forward to receiving your revised manuscript.

Kind regards,

Nnodimele Onuigbo Atulomah, PhD

Academic Editor

Journal Requirements:

1. Please provide additional details regarding participant consent. If you are reporting a retrospective study of medical records, archived samples or third party data, please ensure that you have discussed whether all data were fully anonymized before you accessed them and/or whether the IRB or ethics committee waived the requirement for informed consent. If patients provided informed written consent to have data from their medical records used in research, please include this information

Additional Editor Comments (if provided):

In this study, Bandura’s Social Cognitive Theory has been appropriately selected to provide possible elucidation of the dynamics underpinning the problem phenomenon under study. However, is it not possible that their may be other personal-level dispositions among the population of interest in the study that may play significant role to confirm the researchers’ hypothesis? Additional theoretical considerations may require the inclusion of behavioural control and intention offered by the Theory of Planned Behaviour to confirm or refute the study theoretical position.

As the reviewer 1 observed “Alcohol drinking, smoking cigarettes and tobacco, and marijuana are all illegal substances for adolescents, but there are other influencing factors on drinking, smoking, and marijuana use”. I tend to agree on this point considering that the strength of this study is theoretically grounded in the statement of 5th paragraph of the introduction; “We situated our hypothesis within Albert Bandura’s social learning theory (23). The theorist hypothesised that there are both models and learners in every social learning environment. In this social learning context, learners intentionally or unconsciously observe and imitate behaviours from their models” but weakly operationalized in the data collected. The theory has three constructs; observational learning ( which was operationalized, self-efficacy expectation and outcome expectancy which were absent in the study tended to weaken it

Reviewer 1 has raised some questions needing response “Please explain why alcohol drinking, smoking of tobacco and cigarettes, and marijuana use were grouped together as an outcome variable”

Reviewer 2 has raised a pertinent issue of the recency of data collected in the context of the dynamic nature of behaviour change and the flexibility of selecting variables for the study that may provide contextualization within the theoretical framework.

Reviewers' comments:

Reviewer's Responses to Questions

**Comments to the Author**

1. Does this manuscript meet PLOS Global Public Health’s publication criteria? Is the manuscript technically sound, and do the data support the conclusions? The manuscript must describe methodologically and ethically rigorous research with conclusions that are appropriately drawn based on the data presented.

Reviewer #1: Yes

Reviewer #2: Yes

2. Has the statistical analysis been performed appropriately and rigorously?

Reviewer #1: Yes

Reviewer #2: Yes

3. Have the authors made all data underlying the findings in their manuscript fully available (please refer to the Data Availability Statement at the start of the manuscript PDF file)?

Reviewer #1: Yes

Reviewer #2: Yes

4. Is the manuscript presented in an intelligible fashion and written in standard English?

Reviewer #1: Yes

Reviewer #2: Yes

5. Review Comments to the Author

Reviewer #1: 1. Outcome variable

In this study, the outcome variable was created by combining four single-item questions measuring students' current use of specific substances including cigarettes, marijuana, alcohol and tobacco.

Alcohol drinking, smoking cigarettes and tobacco, and marijuana are all illegal substances for adolescents, but there are other influencing factors on drinking, smoking, and marijuana use.

Please explain why alcohol drinking, smoking of tobacco and cigarettes, and marijuana use were grouped together as an outcome variable.

2. Terminology

In this study, the terminologies are not unified, for instance, 'child,' 'children,' 'child substance use,' 'child current substance use,' 'adolescent substance use,' etc...

Since this study is targeting adolescents, the terminology, 'child' or 'children' (eg, 'child substance use') needs to be unified with 'adolescent' (eg, 'adolescent current substance use') throughout the whole document.

3. Statistics

In Table 2, please check out the Chi-square value and p-value again (probably, one of the p-value or Chi-square-value is wrong).

4. Misspelled

Right under 1.3.3 Multivariable logistic regression, there is typed Table 2 (it will be Table 3).

Reviewer #2: Summary of the research and my overall impression

The study investigated the relationship between exposure to substance and its current usage among school-going adolescents in Timor-Leste country utilizing secondary data from a school-based student health survey conducted in 2015.

Adults and parents were considered as the models, the in-school, adolescents were considered as the learners, while use of the substances was considered as the behaviour.

The authors found that parental tobacco-use and people smoking in the presence of adolescent or second-hand smoking exposure, increased the chance of substance-use among adolescents. Furthermore, being males, adolescents in the upper secondary, and those with food insecurity were more likely to be current substance users whilst adolescents with at least 3 friends and those who experienced parental supervision were less likely to be current users.

Overall the study’s concept is brilliant, and apt for the times, the analysis is thorough and the conclusion are appropriately drawn and reflect the analysis performed. However, the study set out to find information that can guide current practice or strengthen policy direction regarding adolescents’ exposure to substance and it current use in Timor-Leste in the year 2021 and going forward. The information these tasks will rely on for its performance were gathered more than 5 years ago. An original study may well be most appropriate for this type of research question.

Discussion of specific areas for improvement

Major issues:

1. As said in the summary, the data- 2015 utilized for information that is expected to guide or strengthen a current (2021) practice seem relatively old thus making it a significant limitation. This is because behaviour is quite dynamic. Perhaps the scenario should also reflect as a limitation in the study

Minor issues

1. The manuscript at this stage could benefit from line numbering as such a practice makes referencing of queries to become much easier.

2. One other advantage of using original data for this study would be ability freely select control-variables. For example: presence of cigarettes, marijuana, alcohol and tobacco sales’ shops near schools or homes including being sent on errands to buy these substances. This is a comment for future studies in this direction.

Any other points

Confidential comments for the editors

I do not have any other specific concerns about the submission that I’d want the editors to consider before sharing my feedback with the authors.

I have no potentially competing interests, and will be willing to look at a revised version of the manuscript, if available.

6. PLOS authors have the option to publish the peer review history of their article (what does this mean?). If published, this will include your full peer review and any attached files.

**Do you want your identity to be public for this peer review?** For information about this choice, including consent withdrawal, please see our Privacy Policy.

Reviewer #1: No

Reviewer #2: No

---

## [Editor Report · Decision Letter 1]

5 Jan 2022

PGPH-D-21-00643R1Exposure to substance and current substance use of school-going adolescents in Timor-Leste: Evidence from the 2015 Timor-Leste Global school-based student health surveyPLOS Global Public Health

Dear Dr. Ansah,

Thank you for submitting your manuscript to PLOS Global Public Health. As with all papers, your manuscript was reviewed by members of the editorial board. Based on our assessment, we have decided that the work does not meet our criteria for publication and will therefore be rejected.

Specifically:

We are sorry that we cannot be more positive on this occasion. We very much appreciate your wish to present your work in one of PLOS's Open Access publications. Thank you for your support, and we hope that you will consider PLOS Global Public Health for other submissions in the future.

Yours sincerely,

Nnodimele Onuigbo Atulomah, PhD

Academic Editor

Additional Editor Comments (if provided):

This study sought to demonstrate in simple and clear terms that a link thus exist between Exposure to smoking and outcomes of current substance-use experience of in-school adolescents. In the data configurations, the study has not adequately synthesized the variable representing “adolescent Exposure to smoking” adequately and used this with the variable defining current substance-use to compute relative risks and odds ratios for the exposed and unexposed adolescents. This would be a more effective epidemiological risk analysis. The most important data for risk analysis of Exposure to smoking and outcomes of current smoking experience among in-school adolescents are (1) adolescents who reposted being in the presence of an adult who smoked (Exposed) and are currently using substances with addictive potential (Outcome), (2) adolescents who reposted being in the presence of an adult who smoked (Exposed) and are currently NOT using any substance with addictive potential (Outcome), (3) adolescents who reposted have NOT been in the presence of an adult who smoked (NOT Exposed) and are currently using substances with addictive potential (Outcome), (4) adolescents who reposted have NOT been in the presence of an adult who smoked (NOT Exposed) and are currently NOT using any substance with addictive potential (Outcome). This is the core data to demonstrate the claim of the study, from a classical epidemiological perspective. The authors' have responded to the questions of the reviewers adequately, however, these observation emerging from further critical analysis outlined here constitute cause for concern, and are not likely to change the quality of the study since the study has already been conducted.

1) From the critical analysis of data synthesis in this study, there is observed flawed use of data collected to provide support for the study hypothesis.

2) The description of the study design, though stated that source of data was “from low-and-middle-income countries”, Sampling undisclosed number of countries that participated in the study with a total sample size of 4691eligible individuals to participate from 38 schools and 3,704 who actually participated. What was the sample distribution for each country?

3) Measurement weaknesses observed- did not apply the classical measurement techniques for determining epidemiological risk exposure and outcomes for two categories of persons in the study, though the logistic regression modeling was used but the data structure required were no used as explained above; the unexposed who are currently not using any substance of abuse were not factored to determine the risk ratio/odds ratio for clear interpretation.

4) Predicting risk outcomes without recourse to behavior outcomes and antecedents constitute serous oversight in a diagnostic study that requires ecological perspectives identifying important variables in the web of linkages, such as appropriate operationalization of the theories mentioned 1) Social Cognitive Theory and 2) Theory of Planned Behaviour; only the application of these models and theories can effectively provide fully the predictors being sought.

5) Finally, the argument here is beyond risk assessment and diagnostics that characterize this type of study. Bearing in mind that data establishing risk of reinforcing initiation of substance-use by adolescents through secondary smoking experience are replete in literature what is required now this is established is to advance intervention studies that would solve the problem of substance-use by adolescent persist.
---

## [Editor Report · Decision Letter 2]

6 Jul 2022

Exposure to substance and current substance use of school-going adolescents in Timor-Leste: Evidence from the 2015 Timor-Leste Global school-based student health survey

PGPH-D-21-00643R2

Dear %Mr.% %Ansah%,

We are pleased to inform you that your manuscript 'Exposure to substance and current substance use of school-going adolescents in Timor-Leste: Evidence from the 2015 Timor-Leste Global school-based student health survey' has been provisionally accepted for publication in PLOS Global Public Health.

Best regards,

Nnodimele Onuigbo Atulomah, PhD

Academic Editor

All corrections pointed out by the reviewers have been implemented. Congratulations.